# Towards Unlearning Beyond Textual Expressions for LLMs

## Abstract

To remove a designated set of undesirable knowledge from Large Language Models (LLMs), various unlearning approaches have been proposed. Existing approaches typically define the target knowledge through fixed textual expressions and then prevent the model from using it in those specific expressions, catering to only the textual form of the undesirable knowledge, resulting in brittle "forgetting" as unlearnt models may still recover the same knowledge when the expressions are paraphrased or altered. This research thus revisits unlearning in the realistic and failure-prone setting of identifier–attribute (IA) knowledge, where undesirable knowledge cannot be fully captured by fixed expressions. We formalize knowledge extraction under relaxed elicitation conditions by marginalizing over the hidden distribution of query textual expression strategy. This reframes unlearning as minimizing extraction risk over expression variability. Instead of infeasibly sampling over latent prompts, we propose ConRep, a representation-based approach that enforces the invariants implied by the distributional formulation: retains remain stable and surface-invariant, while forgets are repelled and dispersed toward low-information regions in the model's representation space. To evaluate unlearning trustworthily and thoroughly, we build a benchmark ClinicIA, which comprises comprehensive knowledge probing under diverse task formats, spanning unlearning across the settings of two representative knowledge-provenance regimes. Across evaluation tasks and regimes, our approach, ConRep, outperforms prior approaches with remarkable performance with robust forgetting, while preserving the knowledge it should maintain and the LLM's general utility.

## 1 Introduction

Large Language Models (LLMs) memorize vast amounts of sensitive knowledge from training data, raising critical concerns about privacy, copyright, and safety. *Machine unlearning* has emerged as a solution to remove undesirable knowledge while preserving model utility (Cao & Yang, 2015; Si et al., 2023; Liu et al., 2025). Practically, LLM unlearning is operationilized with a set of knowledge designated to forget (i.e. forget set) and another designated to retain (i.e. retain set), an unlearning objective trains model towards minimizing memorization of forget set knowledge while less downgrading memorization of retain set knowledge. However, although with various unlearning objectives designed, existing unlearning approaches largely operate at the textual level, which is fundamentally mismatched to how LLMs manipulate knowledge. For example, supressing LLM from generating the string "Person X died from lung cancer" does not prevent the model from recovering the same fact under rephrased knowledge probes such as "Person X was killd by lung cancer". Since such factual knowledge is acquired from diverse contexts and rephrasings, its latent associations cannot be reliably captured or removed by targeting a single fixed expression.

In this paper, we particularly consider a scenario to unlearn a typical type of knowledge, namely *identifier–attribute* (IA) knowledge, where a fact formed by an entity's identifier (e.g., "John Smith") and corresponding attributes (e.g., "diagnosis": "lung cancer"), as is shown in Figure 1.

We begin by analyzing the shared intuition behind mainstream unlearning objectives. Given the widely adopted view that the essence of a knowledge sample is usually considered as the association between two separable parts [1], unlearning objectives perform to break the association by suppressing

---

[1] e.g. TOFU benchmark (Maini et al., 2024), MUSE benchmark (Shi et al., 2025)

the model's ability to produce the suffix when given the prefix (See Appendix A). This follows an *deterministic view* in which memorization is not directly observable and is therefore operationalized via discoverable knowledge extraction (Nasr et al., 2023; Carlini et al., 2021a), in which, given a fact, a fixed query built from the prefix is posed for the fact and the model is deemed to have "remembered" the fact if it completes with the correct suffix. By this logic, weakening the prefix–suffix link should reduce the risk that a prefix-based probe elicits the suffix. Notably, identifier–attribute (IA) knowledge in common forget/retain sets is often stored as (query, answer) pairs, making this conditional-generation view especially natural .

We argue that this point of view underestimates the empirical risk to be minimized, only accounting for the model's response to a single, fixed expression of a knowledge query, ignoring the diversity of prompts that may reveal the same underlying fact. We thus reformulate extraction as a probabilistic event over a latent distribution of admissible queries, and pose unlearning as minimizing this *distributional extraction risk*. Direct optimization is intractable because the query distribution is latent and sampling is inexhaustive. Our method, **ConRep**, sidesteps this by acting in representation space: we treat a knowledge sample as an anchor whose *semantic neighborhood* implicitly aggregates many admissible expressions. ConRep enforces retain knowledge remains stable across textual variation, while forget knowledge samples are pushed to cluster around a random noise data point in semantically sparse regions, thereby reducing their extractability under any reasonable query.

We introduce a benchmark ClinicIA for Clinical information is a core focus due to its sensitivity and ease of unintended reaccess. To pressure methods under realistic threats, we design probe families based on known fragile triggers in prior probing work—e.g., context-augmented prompts that bypass surface suppression. Evaluation tasks include multiple text generation probes and Multiple Choice Question (MCQ) probes, diversifying knowledge extraction probing with affordable computation. The benchmark includes both pretrained and injected IA knowledge, allowing us to modulate exposure and capture scale-dependent effects. Scoring reflects retain–forget separation across prompt styles, correcting for chance and statistical significance. In aggregate, the benchmark is not just diagnostic: it exposes how easily surface-level forgetting can be undone by minor changes in query or task, and thus sets a falsifiable target for representation-level decoupling.

This work addresses identifier–attribute unlearning through a probabilistic knowledge extraction formulation and a representation-level surrogate. Our contribution is threefold:

**1. Theoretical Reformulation.** We formalize unlearning's core challenge by reframing knowledge extraction as a stochastic event marginalized over the latent distribution of query expressions, moving beyond the deterministic single-template view. This reveals why surface-level forgetting fails: methods optimize for specific textual forms while the underlying knowledge remains accessible through paraphrases and reformulations.

**2. Benchmark - ClinicalIA.** Unlike existing benchmarks that couple evaluation to narrow templates or test only coarse domain removal, ClinicIA approximates the true extraction risk through diverse probe families and carry out complementary knowledge-provenance regimes. By providing statistically meaningful and comparable metrics across methods, it enables the first rigorous assessment of whether unlearning genuinely removes knowledge or merely masks specific expressions.

**3. Unlearning Appoach** We propose ConRep, a representation-based unlearning approach that leverages the insight that semantic neighborhoods in representation space naturally aggregate expression variability, dispersing forget samples toward sparse regions while stabilizing retain samples. Evaluation results on ClinicalIA shows that ConRep achieves robust forgetting that transcends surface-form suppression, as validated by our strongest retain-forget separation results, particularly on expression-variant probes.

## 2 RELATED STUDIES

**LLM memorization & Knowledge Extraction**  Understanding LLM knowledge memorization is fundamental to unlearning formulations. While approaches like linear probing and membership inference can detect memorization, no universal method exists for determining whether knowledge is memorized without supporting assumptions (Chen et al., 2024; Carlini et al., 2019; 2021b; Ishihara, 2023). Practically, a widely adopted surrogate for exact knowledge memorization quantification is knowledge extractability, that is, whether specific knowledge can be retrieved through querying

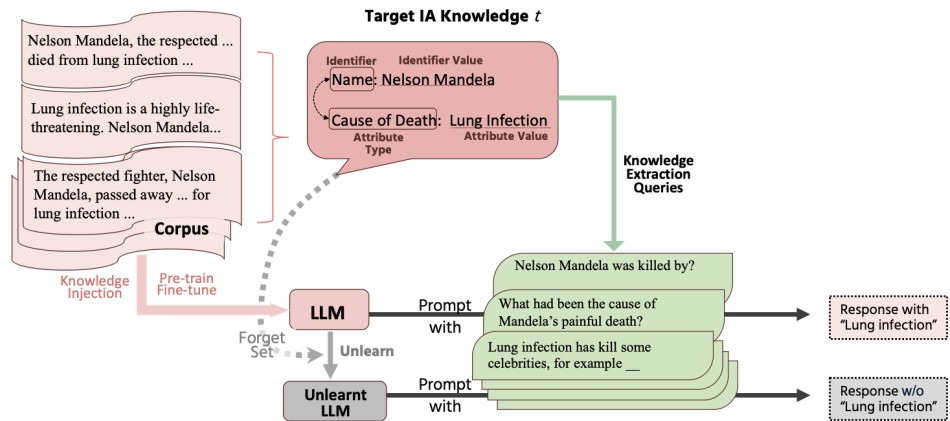

Figure 1: **Illustration of IA unlearning example.** A IA knowledge sample could appear with numerous expressions in the LLM training corpus, leading to the potential of an underlying distribution of queries that may elicit the knowledge. Ideally, the knowledge should not be extracted by queries with any expression.

(Allen-Zhu & Li, 2024). This is quantified via discoverable extraction: splitting training examples into prefix-suffix pairs and testing whether the LLM generates matching suffixes (Hayes et al., 2025; Nasr et al., 2025; Carlini et al., 2021b). However, empirical studies shows that memorization does not guarantee extractability, showing that diverse presentations of the same knowledge offer the basis of its extractability (Allen-Zhu & Li, 2024; Wang et al., 2025) . This creates fundamental uncertainty about which queries can successfully extract specific knowledge, as the space of effective prompts is determined by the training-time exposure patterns and cannot be exhaustively enumerated.

Most unlearning approaches are framed by how they view probing of memorized knowledge, while most adopt deterministic discoverable extraction (Nasr et al., 2025). However, this yields a yes-or-no determination of whether extraction was successful with respect to a single user query, most typically executed with deterministic greedy sampling, empirical studies from (Hayes et al., 2025) addressed this underestimates extraction risks by a notable margin.

**Unlearning Evaluation** Existing benchmarks differ in evaluation task formulation, but none explicitly targets identifier–attribute unlearning, thus cannot reliably rank or select methods for such cases. WMDP (Li et al., 2024), targeting harmful domain knowledge (e.g., biosecurity, chemical weapons), uses multiple-choice questions as a knowledge manipulation test; this provides controlled measurement but cannot verify actual forgetting of generative knowledge. Who's Harry Potter (Eldan & Russinovich, 2024) evaluates domain-level forgetting on fictional content through model-generated outputs, yet its informal setup makes distinguishing suppression from true forgetting unclear. TOFU (Maini et al., 2024) initially formulates knowledge explicitly as synthetic identifier–attribute profiles, but in evaluation, adopts a QA-based knowledge extraction format. Considering insights from memorization studies (Hayes et al., 2025; Nasr et al., 2025), such QA formulations might underestimate the true extraction risk, as the originally structured identifier–attribute knowledge could still be extractable via diverse prompting methods beyond the given QA tasks.

Recent critiques argue about these benchmarks for their insufficiency in evaluating model knowledge preservation after unlearning. (Thaker et al., 2025) show that small prompt changes can recover supposedly forgotten knowledge, questioning benchmark generalization. (Hu et al., 2025) demonstrate benign fine-tuning can restore erased content, revealing latent persistence not captured by test metrics. (Zheng et al.) find that wrong answers may reflect misalignment rather than erasure, showing that behavioral success can be spurious. Overall, current benchmarks lack robustness to probe-space variation and cannot verify whether unlearning actually disentangles knowledge internally.

**Approaches for LLM Unlearning** Mainstream preexisting unlearning approaches could be divided into two categories: LM-Loss-based approaches that directly manipulate the language modeling objective (Liu et al., 2025), and representation-based approaches that intervene in the model's internal representations during knowledge processing.

*LM-loss-based approaches* directly optimize generation probabilities for target content. Gradient Ascent (GA) (Jang et al., 2023) pioneers in applying gradient ascent on forget sequences to maximize negative log-likelihood, Zhang et al. (2024) proposed Negative Preference Optimization (NPO) on its basis to prevent catastrophic collapse. These methods unlearn knowledge as their fixed prompt templates, regardless diversity of possible expression, and show fragility in maintaining consistent forget rates when prompts are rephrased or restructured, even on benchmarks where they report impressive results (Thaker et al., 2025; Hu et al., 2025; Zheng et al.). *Representation-based approaches* manipulate internal model states to achieve forgetting. Representation Misdirection for Unlearning (RMU) (Li et al., 2024; Dang et al., 2025) steers forget representations toward random directions, while mechanistic approaches (Guo et al., 2025) use interpretability techniques to localize and target specific factual recall circuits within transformer architectures. Although transcending word-by-word unlearning limitations through representation-level interventions that capture abstract semantics, RMU is primarily designed for coarse-grained domain knowledge removal (Li et al., 2024; Dang et al., 2025), the performance in fine-grained knowledge forgetting, such as specific identifier-attribute relationships in TOFU, remains limited (Dorna et al., 2025; Thaker et al., 2025).

Given these limitations, there is a clear need for unlearning approaches that can both transcend rigid textual expressions and achieve fine-grained control over knowledge removal, particularly for identifier-attribute unlearning scenarios.

## 3 PROBLEM FORMULATION

### 3.1 PRELIMINARIES: DETERMINISTIC EXTRACTION

Most unlearning methods work on forget/retain sets using losses defined at *fixed* query templates. Let $\mathcal{F}$ and $\mathcal{K}$ denote forget and retain sets. An IA fact $t = (I, p, v)$—$I$ is an identifier (such as Nelson Mandela, $p$ is an attribute type (such as Cause of Death, and $v$ is an attribute value (such as Lung Infection)—is included in either $\mathcal{F}$ or $\mathcal{K}$. Given $t$, a text generaton model $g$ builts a query $q$ from a fixed textual expressoin template $s_0$: $q = g(I, p; s_0)$. A judge function $J(a, v)$ returns 1 if given answer $a$ to query $q$ and the fact $v \in t$ are identical; Otherwise 0.

In general, A common objective function in LLM unlearning can be formulated as follows,

$$\min_{\theta} \underbrace{\frac{1}{|\mathcal{F}|} \sum_{t \in \mathcal{F}} \ell_{\text{forget}}\big(M, g(I, p; s_0), v\big)}_{\text{lower success under the fixed query}} + \lambda \underbrace{\frac{1}{|\mathcal{K}|} \sum_{t \in \mathcal{K}} \ell_{\text{retain}}\big(M, g(I, p; s_0), v\big)}_{\text{preserve success under the fixed query}}, \quad (1)$$

where $M$ is the target LLM for unlearning. LM-loss methods instantiate $\ell_{\text{forget}}$ via (negative) LM likelihoods (e.g., GA/NPO variants), while representation methods pull/push hidden states but still *measure* success at fixed $s_0$ (or a tiny template set $\mathcal{S}_0$). Thus the underlying surrogate is a point (or few-point) estimate tied to surface text.

We refer to succesful indentification of $v$ given $I$ and $p$ as *extraction*. The deterministic extraction found in past work can be forumated as

$$P_{\text{det}}(v|I, p; M, g, s_0) = J\big(M(g(I, p; s_0)), v\big).$$

### 3.2 STOCHASTIC EXTRACTION: A GENERALIZED FORMULATION

For IA facts—acquired across diverse contexts and phrasings—the deterministic extraction is a *degenerate* probe: minor paraphrases or backgrounding often revive "forgotten" facts. To align with this reality, we re-define extraction as success *marginalized* over prompt variability randomness. Let $s$ be drawn from $\mathcal{S}$ a distribution over admissible various IA queries. We define the stochastic extraction as

$$P_{\text{sto}}(v|I, p; M, g, \mathcal{S}) = \mathbb{E}_{s \sim \mathcal{S}}\big[J\big(M(g(I, p; s)), v\big)\big]. \quad (2)$$

Deterministic extraction is the special case with $\mathcal{S} = \{s_0\}$.

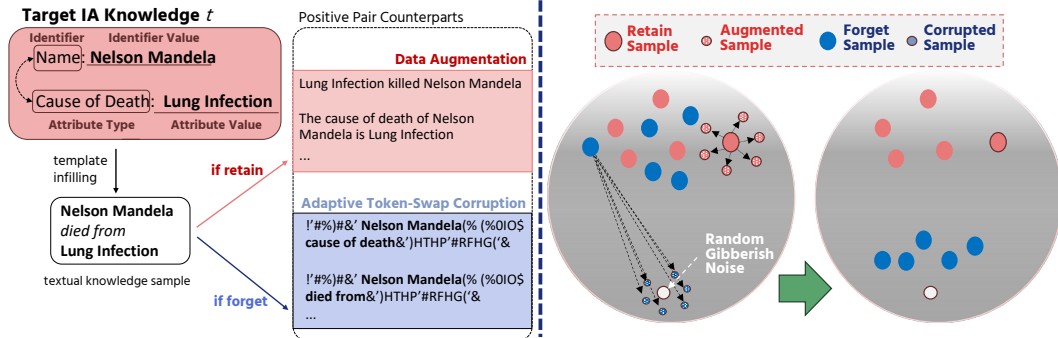

Figure 2: **Representation-space intuition of our unlearning approach.** The left side shows an example of how we obtain the positive pair counterparts for implementing retain/forget contrastive loss; the right side illustrates the intuition of our objective design, i.e., to let the retain sample stay and stabilize, and push forget samples to cluster around a random gibberish noise.

## 3.3 Unlearning Objective under Probabilistic Extraction

We seek low extraction on the forget set and high extraction on the retain set:

$$\frac{1}{|\mathcal{F}|}\sum_{t\in\mathcal{F}} P_{\text{sto}}(v|I,p;M',g,\mathcal{S}) \leq \varepsilon, \qquad \frac{1}{|\mathcal{K}|}\sum_{t\in\mathcal{K}} P_{\text{sto}}(v|I,p;M',g,\mathcal{S}) \geq \sigma, \tag{3}$$

for a post-unlearning model $M'$ and thresholds $\varepsilon \ll \sigma$. Therefore, we minimize an objective:

$$\mathcal{J}(M';\mathcal{F},\mathcal{K},g,S) = \underbrace{\frac{1}{|\mathcal{F}|}\sum_{t\in\mathcal{F}} P_{\text{sto}}(v|I,p;M',g,\mathcal{S})}_{\text{forget risk}} - \lambda \underbrace{\frac{1}{|\mathcal{K}|}\sum_{t\in\mathcal{K}} P_{\text{sto}}(v|I,p;M',g,\mathcal{S})}_{\text{retain utility}}, \quad \lambda > 0.$$

$$\tag{4}$$

## 4 Unlearning Approach: ConRep

Directly minimizing $\mathcal{J}$ is infeasible because of the inexhaustible sampling of the underlying admissible query $\mathcal{S}$ distribution. Inspired by representation-steering studies (Zhao et al., 2025; Singh et al., 2024) that seek fine-grained control of LLMs in the concept level that break the limitation of text, we introduce a *representation-level surrogate* that approximates the distribution of knowledge query-answer pairs as a neighborhood reflecting knowledge semantics in the LLM's embedding space, and seek to remove designated knowledge while preserving other by reshaping the geometry of the model's representation space driven by contrastive loss (Wu et al., 2022; Rimsky et al., 2024). The intuitive illustration of our approach is shown in Figure 2

### 4.1 Basic Unlearning Objective Design

**Preliminaries.** Let $x$ be a sequence and $\boldsymbol{h}(x) \in \mathbb{R}^d$ the pooled final-layer representation (mean-pooling and $\ell_2$-normalization by default). We build our method with contrastive learning to directly "pull" or "push" points in this space at the *semantic* level. We consider disjoint sets $\mathcal{R}$ (retain set) and $\mathcal{F}$ (forget set). For each retain sample $r \in \mathcal{R}$, we have $K$ textual augmentations $\{\tilde{r}^{(k)}\}_{k=1}^{K}$ (see Appendix B). For each forget example $f \in \mathcal{F}$, we construct $V$ noisy views $\{c^{(i)}(f)\}_{i=1}^{V}$.

**Retain contrastive loss.** We encourage retain samples to align in the space to cluster and push away forget samples together, while taking augmentation of the pivot as hard-positive to stablize the semantics of the pivot sample:

$$\mathcal{L}_{\text{gen}}^{\text{ret}} = -\frac{1}{|\mathcal{P}|}\sum_{r\in\mathcal{P}} \log \frac{\sum_{u\in\mathcal{A}(r)\cup\mathcal{R}\setminus\{r\}} \exp\big(\langle\boldsymbol{h}(r),\boldsymbol{h}(u)\rangle/\tau_r\big)}{\sum_{z\in\mathcal{R}\cup\mathcal{F}} \exp\big(\langle\boldsymbol{h}(r),\boldsymbol{h}(z)\rangle/\tau_r\big)}, \tag{5}$$

where $\mathcal{P}$ is the set of retain pivots, $\mathcal{A}(r)$ the augmentations of $r$, and $\tau_r$ the retain temperature. This *general* positive set (pivot's own augmentations *plus* in-batch retain) serves broad utility preservation in non-IA-structured or distribution-mismatched retain scenarios.

**Forget contrastive loss.** We destroy targeted IA binding for forget samples by push their representation in semantic space toward random noise, while (optionally) allowing weak cohesion among forgets samples:

$$\mathcal{L}_{\text{gen}}^{\text{forg}} = -\frac{1}{|\mathcal{F}|} \sum_{f \in \mathcal{F}} \log \frac{\sum_{i=1}^{V} \exp\big(\langle \boldsymbol{h}(f), \boldsymbol{h}(c^{(i)}(f)) \rangle / \tau_f\big) \; + \sum_{f' \in \mathcal{F} \setminus \{f\}} \exp\big(\langle \boldsymbol{h}(f), \boldsymbol{h}(f') \rangle / \tau_f\big)}{\sum_{z \in \{c^{(i)}(f)\}_{i=1}^{V} \cup \mathcal{F} \cup \mathcal{R}} \exp\big(\langle \boldsymbol{h}(f), \boldsymbol{h}(z) \rangle / \tau_f\big)} . \quad (6)$$

Here $\tau_f$ denotes the temperature hyperparameter.

*Constructing a Random-Noise Neighborhood.* A naive single global noise vector would collapse all forget anchors to (nearly) the same point; a purely random noise vector brings risk for controbility (e.g. it may fall in somewhere dense retain samples cluster). We therefore construct, for each forget anchor $f$, an *random-noise neighborhood* via **token-swap corruption**:

$$c^{(i)}(f) \; = \; \text{Corrupt}_\rho\big(f, n^{(i)}, \omega^{(i)}\big), \quad (7)$$

where we probabilistically replace tokens in $f$ with corresponding positions from a random sequence $n^{(i)}$ according to mask $\omega^{(i)}$ with rate $\rho$. This strategy ensures that: (i) the resulting representations remain within the model's semantic space (avoiding optimization instability), (ii) each forget sample disperses to its own neighborhood rather than converging to a shared point. The outcome is a semantically impoverished region that is reachable during optimization yet ineffective for IA recall.

**Specializing Symmetric IA Unlearning Objective.** In our preliminary experiments , we found that when both retain and forget sets are IA structured and sampled from similar distributions (e.g., different slices of medical IA), the distribution-agnostic objectives in equation 5 and equation 6 expose two failure modes: (i) treating any in-batch retain as positives compresses distinct identifiers, harming IA separability; (ii) letting forgets be mutual positives creates a single forget block that remains extractable under prompt variation. Motivated by this, we derive special setups for constructing positives/negatives under pairs under a symmetric setting, details are shown in subsection B.1.

**Final loss.** With the retain and forget contrastive loss to guide reshaping the geometry of the model's representation space, we add the final total unlearning objective with an auxiliary Language Model (LM) loss term to discourage the model from rebuilding fluent predictions on forget inputs while gently adapting the LM head to retain inputs (via $\gamma$), improving instruction-following stability:

$$\mathcal{L}_{\text{con}} = w_f \, \mathcal{L}^{\text{forg}} \; + \; (1 - w_f) \, \mathcal{L}^{\text{ret}}, \quad (8)$$

$$\mathcal{L}_{\text{LM}} = \mathbb{E}_{x \in \mathcal{F}}\big[\text{NLL}_\theta(x)\big] \; - \; \gamma \, \mathbb{E}_{y \in \mathcal{R}}\big[\text{NLL}_\theta(y)\big], \quad (9)$$

$$\mathcal{L}_{\text{total}} = (1 - \lambda_{\text{LM}}) \, \mathcal{L}_{\text{con}} \; - \; \lambda_{\text{LM}} \, \mathcal{L}_{\text{LM}}. \quad (10)$$

# 5 BENCHMARK: CLINICIA

We introduce a new benchmark **ClinicIA** (**Clicic**al **I**dentifier-**A**ttribute Unlearning) to provide trustworthy evaluation for IA unlearning faced with the limitation of pre-existing unlearning benchmarks: they couple evaluation to textual forms (fixed prompt or a narrow template set) and underestes extraction risk under paraphrases or alternative evaluation task formats.

ClinicIA closes this gap by approximating the latent query-expression distribution with diversified **evaluation tasks** and by evaluating across complementary knowledge-provenance **regimes**. These choices yield a stricter, distribution-aware assessment of IA unlearning while remaining practical.

We ground this benchmark in clinical IA knowledge, for both its real-world salience to trustworthy medical deployment, and its vulnerability under expression variability, since highly structured clinical IA knowledge is nevertheless learned from highly diverse training setups [2]. To enable unlearning for all approaches, the structured IA knowledge samples are to declarative sentences for training.

---

[2]For example, Next-token-prediction training on clinical notes, medical dialogues, articles, etc.

**Evaluation tasks**. To comprehensively assess unlearning robustness, we employ two complementary probe families that test different aspects of knowledge retention: *generation probes* evaluate the model's ability to produce knowledge, while multiple-choice question (*MCQ*) probes test recognition without requiring generation, together capturing both active recall and passive identification.

*Generation Probes*. Since unlearning operates on declarative statements while knowledge can be elicited through diverse query forms, we probe via three expression variants: (i) QA probes, direct questions about the IA fact, (ii) Cloze probes, fill-in-the-blank templates, and (iii) Background-augmented probes— , queries preceded by contextual information about the entity. The first two represent standard rephrasings that approximate the distribution of natural queries, while background-augmentation specifically targets a known vulnerability where contextual priming can bypass surface-level suppression, as demonstrated in recent unlearning critiques. We denote them as *Q/C/B* for remaining part respectively.

*MCQ Probes*. Adopt multiple-choice format to test knowledge extraction. Each query presents four options, and the model must choose the correct one. The confidence that LLM hold of each answer is evaluated based on the log-likelihood of each option, with the highest choice considered as the predicted answer. This probe family includes three types: (i) Identifier-Equal (ID-eql)—selecting the correct identifier given an attribute value, (ii) Identifier-Closest (ID)—finding the identifier with the nearest attribute value, and (iii) Attribute-Equal (ATT)—identifying the correct attribute for a given identifier. Altoghther, they tests whether knowledge remains accessible even when generation is successfully suppressed. We denote them as *IDeq/ID/ATT* respectively throughout remaining part.

**Datasets and regimes.** We construct ClinicIA with two knowledge-provenance regimes to systematically evaluate unlearning along two critical dimensions: (1) knowledge embedding depth—whether IAs are deeply embedded from pretraining or newly injected via finetuning, and (2) retain-forget distributional symmetry—whether retain and forget sets come from distinct or identical distributions.

*Regime A: Pre-embedded with weak symmetry.* Celebrity deaths and diagnoses embedded during pretraining test unlearning of deeply ingrained knowledge. We forget one celebrity IA type (e.g., deaths) while retaining general Wikipedia knowledge, but crucially evaluate on both Wikipedia and the complementary celebrity type (e.g., diagnoses). This "weakly symmetric" design—asymmetric training but symmetric evaluation—tests whether methods can precisely remove specific IAs while preserving structurally similar ones, challenging unlearning when knowledge is deeply rooted.

*Regime B: Injected with strong symmetry.* Synthetic clinical IAs from PMC-Patients enable controlled injection where retain/forget sets are randomly split from identical distributions. Each IA appears across clinical notes, dialogues, and articles, creating rich expression diversity. setting. In this case retain/forget could be considered i.i.d, we note as "strongly symmetric" senario, representing the hardest precision challenge, testing whether methods can achieve selective forgetting without distributional cues.

These complementary regimes reveal whether methods fail due to insufficient forgetting (Regime A: deep embeddings resist removal) or imprecise forgetting (Regime B: symmetric distributions cause collateral damage). Details about datasets and regime settings see Appendix C.1.1 and C.1.2.

**Evaluation Metrics.** To make results comparable across probes of different difficulty, we report baseline-normalized scores based on both the accuracy of the baseline model (no unlearning) and the unlearnt model. We also flip the forgetting axis so that "more forgetting" maps to larger values to keeps directions consistent with "better retention". For MCQ we apply chance correction so that random guessing evaluates near zero rather than inflating results, and we use significance filtering to hide normalized MCQ scores when the baseline is not reliably above chance, avoiding unstable ratios. Exact formulas and statistical tests are provided in Appendix C.2

# 6 EXPERIMENTAL RESULTS

## 6.1 EXPERIMENT SETUP

**Models.** We implement unlearning on ClinicIA with Llama-2-7b-chat-hf (Touvron et al., 2023), and Mistral-7B-Instruct-v0.2, two open-source LLMs with advanced knowledge manipulation ability at their sizes (Jiang et al., 2023) that have been widely used.

Table 1: **Regime A (Unlearnt on Diagnosis).** Baseline shows raw accuracies; others show relative scores. MMLU is raw and excluded from Avg.  [†] not significant

| | | Retain | | Forget | | Avg. |
|---|---|---|---|---|---|---|
| **Model** | **MMLU** | **Death Generation** Q/C/B ↑ | **Death MCQ** IDeq/ATT/ID∼ ↑ | **Diagnosis Generation** Q/C/B ↓ | **Diagnosis MCQ** ID/ATT ↓ | |
| **Llama2** | | | | | | |
| Baseline (Acc.) | 0.464 | 0.53/0.52/0.25 | 0.29/0.39/0.34 | 0.23/0.10/0.13 | 0.29[†]/0.51 | |
| + Graddiff | 0.461 | 92.5/100.0/100.0 | 100.0/87.9/90.4 | 33.4/60.0/0.00 | –/30.2 | 69.4 |
| + NPO | 0.463 | 97.5/99.2/100.0 | 100.0/100.0/90.4 | 25.0/0.0/28.5 | –/0.0 | 64.1 |
| + RMU | 0.462 | 100.0/100.0/100.0 | 77.7/94.0/100.0 | 33.4/60.0/0.0 | –/83.0 | 74.8 |
| **+ ConRep** | 0.455 | 95.8 / 90.7 / 75.9 | **100.0**/87.9/**100.0** | **91.7**/20.1/**42.9** | –/67.9 | **77.3** |
| **Mistral** | | | | | | |
| Baseline (Acc.) | 0.590 | 0.64/0.71/0.58 | 0.29/0.59/0.25[†] | 0.19/0.21/0.42 | 0.35[†]/0.55 | |
| + Graddiff | 0.517 | 0.0/0.0/0.0 | 0.0/51.3/– | 100.0/100.0/100.0 | –/98.4 | 50.0 |
| + NPO | 0.574 | 15.7/48.8/8.3 | 100.0/74.4/– | 100.0/100.0/95.5 | –/100.0 (+11.5) | 71.4 |
| + RMU | 0.583 | 91.8/91.4/57.1 | 100.0/73.1/– | 100.0/100.0/54.5 | –/100.0 (+44.3) | 85.3 |
| **+ ConRep** | **0.583** | **91.8**/84.0/**64.7** | **100.0**/ 97.4 /– | **100.0**/81.8/86.4 | **–/100.0** (+11.5) | **89.6** |

Table 2: **Regime B.** Baseline row reports accuracy to each . Other rows are percentages relative to baseline; MCQ uses chance correction with $p_0 = 0.25$.  [†] not significant

| Model | MMLU | **Generation** (Q/C/B) | **Δ Gen** (F%−R%) | **MCQ** (ATT/IDeq/IDident) | **Δ MCQ** (F%−R%) |
|---|---|---|---|---|---|
| **Mistral** | | | | | |
| Baseline | 0.269 | 0.567/0.742/0.325 | – | 0.374/0.320[†]/0.280[†] | – |
| + Graddiff | 0.230 | **R%**: 116.4/90.7/166.5 **F%**: 111.1/86.3/156.9 | -5.3/-4.4/-9.5 | **R%**: 8.1/–/– **F%**: -8.1/–/– | -16.1/–/– |
| + NPO | 0.231 | **R%**: 98.1/90.3/137.5 **F%**: 93.5/87.6/132.3 | -4.6/-2.7/-5.2 | **R%**: 32.3/–/– **F%**: 201.6/–/– | 169.4/–/– |
| + RMU | 0.272 | **R%**: 96.6/89.9/150.2 **F%**: 91.7/86.3/153.8 | -4.9/-3.6/ 3.7 | **R%**: 80.6/–/– **F%**: 8.1/–/– | -72.6/–/– |
| **+ ConRep** | **0.266** | **R%**: 97.4/99.5/165.8 **F%**: 88.2/87.6/144.6 | **-9.2/-11.9/-21.2** | **R%**: 201.6/–/– **F%**: 185.5/–/– | -16.1/–/– |

**Utility Metric.** We use MMLU (Massive Multitask Language Understanding) (Hendrycks et al., 2021),the widely adopted evaluation protocal for model's general knowledge manipulation ability, to assess if (or how much) the unlearning hurt LLM's general utility.

**Compared Approaches** We compare our proposed **ConRep** with three competitive unlearning approaches: Graddiff (Yao et al., 2024), NPO (Zhang et al., 2024), RMU (Li et al., 2024).

**Training Details** We set default epoch number to be 10 epochs and early stpos when the drop of utility greater than gain of knowledge probe accuracy. For Regime A to unlearn Pre-embedded IA knowledge, we operate directly on the released LLMs checkpoints; For Regime B to unlearn post-hoc injected IA knowledge, we finetune LLMs with the language modelling task on both the forget set and retain set, and then implement unlearning for the finetuned model. As llama-2-7b-chat-hf scores a low performance result even before unlearning, we consider it deficient to memorize the involved synthetic clinical IA knowledge concretely, and only report performance on Mistral-7B-Instruct-v0.2 for regime B. Further implementation details see Appendix E.

## 6.2 RESULTS AND ANALYSIS

We evaluate unlearning performance across our two complementary regimes, examining how methods handle different embedding depths and symmetry levels. Table 1 shows Regime A results with

forget set being Celebrity Diagnosis; We also report performance in the complementary setup (forget set being Celebrity Deaths) in Appendix D.3.1.

**Regime A: Deep embeddings expose surface-level brittleness.** When unlearning pre-embedded celebrity IAs, the performance patterns directly validate our benchmark's design principles. On Llama-2, ConRep achieves the highest forget scores on Diagnosis generation (91.7/20.1/42.9 for Q/C/B), with the dramatic difference between Q (91.7) and C (20.1) probes revealing how even simple template variations can resurrect supposedly "forgotten" knowledge in surface methods. The background-augmented probe (B) serves as our most stringent test—here ConRep's 42.9 vastly exceeds competitors (all scoring 0.0), demonstrating that contextual priming bypasses textual suppression but not representation-level decoupling.

The Mistral results reveal a ceiling effect where all methods achieve near-perfect generation forgetting, yet differentiation emerges through utility preservation: ConRep maintains the highest MMLU (0.583 vs 0.517 for Graddiff), confirming that representation reorganization with LM guidance avoids the catastrophic interference plaguing gradient-based methods. This pattern—where aggressive forgetting damages general capabilities—underscores why operating in representation space is crucial for practical deployment.

**Regime B: Controlled injection with strong symmetry.** Table 2 exposes a fundamental challenge in symmetric unlearning: when retain and forget sets share identical distributions, augmentation benefits both sides. All methods show $R\%$ and $F\%$ exceeding baseline, creating an apparent paradox where "better" performance occurs on both retain and forget, indicating the tough knowledge injection with compromising on general utility (low baseline MMLU performance) may lead to an unstable knowledge manipulation scenario, but an advanced unlearning method could still make a difference. Our ConRep's $\Delta$ values (-9.2/-11.9/-21.2) reveal the critical insight: what matters is not absolute performance but the differential gap. The -21.2 on background-augmented probes—nearly 6× larger than competitors—demonstrates that semantic decoupling in representation space maintains separation even when surface-level distinctions vanish.

**Cross-regime insights.** Two patterns emerge across regimes: (1) **Expression robustness**—ConRep consistently excels on Cloze and Background-augmented probes, which test resistance to paraphrasing and contextual priming, confirming that semantic-level intervention transcends template-specific suppression. (2) **Utility preservation**—Unlike Graddiff which damages MMLU severely, ConRep maintains baseline utility (Regime A: 0.455 vs 0.464; Regime B: 0.266 vs 0.269), demonstrating that representation reorganization with LM guidance preserves general capabilities while achieving targeted forgetting.

## 7 CONCLUSION

This work identifies and addresses a fundamental gap in LLM unlearning: the mismatch between how knowledge is encoded (across diverse expressions) and how it is targeted for removal (via fixed templates). With reformalizing unlearning objective, we stress why existing methods fail: they optimize for specific textual forms while knowledge remains accessible through unlimited variations.

Our contributions span formulation, unlearning approach, and benchmarking. The ConRep method demonstrates that operating in representation space—where semantic neighborhoods naturally aggregate expression variability—achieves robust forgetting that withstands prompt perturbations. The ClinicIA benchmark reveals this superiority empirically: across deep pre-embedded knowledge and symmetrically distributed injected knowledge, ConRep consistently achieves the strongest retain-forget separation, particularly on expression-variant probes where surface methods fail.

These findings establish that robust IA unlearning requires intervention at the semantic level rather than output suppression. While challenges remain—including model-specific variations and scalability to larger knowledge sets—this work provides both theoretical grounding and practical validation for achieving true knowledge removal. Future directions include mechanistic analysis of representation changes, extension to other knowledge types beyond IA, and integration with constitutional AI frameworks for safer deployment.

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

## A    LLM MEMORIZATION QUANTIFICATION: KNOWLEDGE EXTRACTION

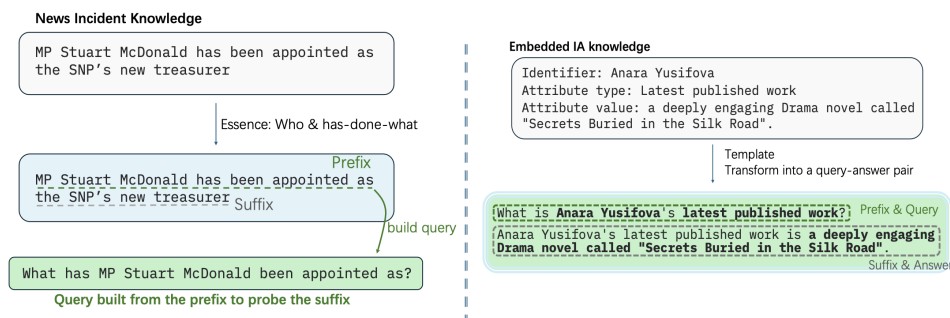

Figure 3: Left: from MUSE benchmark, news incident knowledge; right: from TOFU benchmark, IA knowledge

This appendix illustrates the fundamental prefix-suffix paradigm that underlies mainstream unlearning approaches and exposes its limitations for IA knowledge. As shown in Figure 3, existing unlearning methods conceptualize knowledge as decomposable into *prefix* and *suffix* components, where the association between these parts constitutes the essence of the knowledge. This decomposition manifests differently across knowledge types:

**News Incident Knowledge (MUSE**[3]**)**: The prefix contains the contextual setup ("MP Stuart McDonald has been appointed as the SNP's new treasurer"), while the suffix holds the completion. Unlearning targets the model's ability to complete this specific textual pattern.

**Identifier-Attribute Knowledge (TOFU**[4]**)**: The knowledge is structured as identifier-attribute pairs, where queries probe specific attributes ("What is Anara Yusifova's latest published work?"). The prefix encompasses the identifier and attribute type, while the suffix contains the attribute value.

The figure reveals a critical disconnect between how unlearning is *implemented* (blue annotations) and how it is *evaluated* (green annotations):

**Training Phase (Blue)**: Unlearning objectives directly manipulate the prefix-suffix co-occurrence probability, typically by maximizing the negative log-likelihood of generating the suffix given the prefix. This assumes that breaking this single textual link will prevent knowledge extraction.

**Evaluation Phase (Green)**: Knowledge extraction is tested through queries built from the prefix to probe whether the model can still produce the suffix. However, evaluation typically uses only the *same fixed query template* used during training.

## B    METHODOLOGY DESIGN DETAILS

### B.1    TRICK TO ENHANCE SYMMETRIC IA UNLEARNING

**Trick 1: Pivot-only positives for retain.**    Let $\mathcal{R}$ be all retain embeddings and $\mathcal{F}$ all forget embeddings. For each retain pivot $r$ with its augmentation set $\mathcal{A}(r)$, only $\mathcal{A}(r)$ are positives; *all* other retain (other pivots and their augments) and *all* forget are negatives:

$$\mathcal{L}_{\text{sym}}^{\text{ret}} = -\frac{1}{|\mathcal{P}|} \sum_{r \in \mathcal{P}} \log \frac{\sum\limits_{u \in \mathcal{A}(r)} \exp\big(\langle \mathbf{h}(r), \mathbf{h}(u) \rangle / \tau_r\big)}{\sum\limits_{z \in \mathcal{R} \cup \mathcal{F}} \exp\big(\langle \mathbf{h}(r), \mathbf{h}(z) \rangle / \tau_r\big)}. \tag{11}$$

This preserves IA separability across identifiers while clustering each identifier's expressions.

**Trick 2: Per-anchor noisy positives for forget (no mutual positives) with retain-push.**    For each forget anchor $f$, only its own token-swap noisy views $\{c^{(v)}(f)\}_{v=1}^{V}$ are positives; all other forgets

---

[3]https://muse-bench.github.io/

[4]https://locuslab.github.io/tofu/

and all retain are negatives. We push away from retain via a negative-column weight $w_{\mathrm{ret}} \geq 1$ and an additive margin $m \geq 0$ on retain logits. Define the compact sums

$$s_f^+ = \sum_{v=1}^{V} \exp\left( \frac{\langle \mathbf{h}(f), \mathbf{h}(c^{(v)}(f)) \rangle}{\tau_f} \right), \; s_f^{F-} = \sum_{f' \in \mathcal{F} \setminus \{f\}} \exp\left( \frac{\langle \mathbf{h}(f), \mathbf{h}(f') \rangle}{\tau_f} \right), \; s_f^{R-} = \sum_{r \in \mathcal{R}} \exp\left( \frac{\langle \mathbf{h}(f), \mathbf{h}(r) \rangle + m}{\tau_f} \right),$$

(12)

we have the loss defined as

$$\mathcal{L}_{\mathrm{sym}}^{\mathrm{forg}} = -\frac{1}{|\mathcal{F}|} \sum_{f \in \mathcal{F}} \log \frac{s_f^+}{s_f^+ + s_f^{F-} + w_{\mathrm{ret}} \, s_f^{R-}}.$$

(13)

Temperature separation $\tau_f < \tau_r$ further sharpens forget dispersion while keeping retain cohesion smooth.

**Relation to the general case objective.** Compared with general case objectives shown by equation 5 and equation 6, the symmetric specialization changes only the *positive/negative set design* (pivot-only and per-anchor) and adds a *retain-push* (weight+margin) for forget. All other components (on-manifold token-swap control views, per-anchor multi-view construction, and temperature split) remain shared with the general form.

## C CLINICIA BENCHMARK DETAILS

### C.1 DATASETS

#### C.1.1 REGIME A: PRE-EMBEDDED CELEBRITY IAS

- **Celebrity Deaths**: 450 public figures' cause and year of death from Wikipedia/Kaggle[5]
- **Celebrity Diagnosis**: 58 celebrities' medical conditions from MedPage Today, cross-verified with Wikipedia[6]
- **Training configuration**: Due to limited dataset sizes (¡100 samples), we use general Wikipedia passages as retain set to avoid overfitting, while reserving the complementary celebrity dataset for evaluation
- **Data format**: Converted to declarative statements for training (e.g., "Nelson Mandela died from lung infection")

#### C.1.2 REGIME B: INJECTED CLINICAL IAS

- **PMC Clinical IAs**: 25,000 synthetic patient records with attributes (age, gender, diagnosis)
- **Source materials**: Asclepius medical dialogues (Kweon et al., 2024) and PMC-Patients clinical summaries (Zhao et al., 2022)
- **Expression contexts**: Each IA embedded in clinical discharge notes, medical dialogue transcripts, and research article excerpts
- **Current experiments**: Use highest-diversity setting; lower-diversity variants reserved for future model capacity studies
- **Train/test split**: 900 retain / 100 forget after quality filtering

### C.2 EVALUATION METHODOLOGY

#### C.2.1 SCORE NORMALIZATION

We employ baseline-relative, chance-corrected scoring to ensure comparability across probes of different difficulty and to maintain directional consistency between retain and forget metrics.

---

[5]https://www.kaggle.com/datasets/hugodarwood/celebrity-deaths
[6]https://www.medpagetoday.com/popmedicine/celebritydiagnosis

**Generation Probes (Q/C/B).** For unlearned model accuracy $S_{\text{unl}}$ and baseline accuracy $S_{\text{base}}$:

$$\text{Retain score} = \text{clip}_{[0,100]}\left(100 \times \frac{S_{\text{unl}}}{S_{\text{base}}}\right); \tag{14}$$

$$\text{Forget score} = \text{clip}_{[0,100]}\left(100 \times \left(1 - \frac{S_{\text{unl}}}{S_{\text{base}}}\right)\right). \tag{15}$$

**MCQ Probes (chance-corrected).** With chance probability $p_0 = 0.25$:

$$\text{Retain score} = \text{clip}_{[0,100]}\left(100 \times \frac{S_{\text{unl}} - p_0}{S_{\text{base}} - p_0}\right); \tag{16}$$

$$\text{Forget score} = \text{clip}_{[0,100]}\left(100 \times \left(1 - \frac{S_{\text{unl}} - p_0}{S_{\text{base}} - p_0}\right)\right). \tag{17}$$

Clipping to [0,100] prevents rewards for exceeding baseline ($> 100$) and over-penalization for benign performance dips ($< 0$), ensuring scores reflect our intended evaluation goals.

### C.2.2 STATISTICAL SIGNIFICANCE TESTING

For MCQ probes, we perform significance testing to avoid unstable normalization when baseline performance is not reliably above chance. We test the null hypothesis $H_0 : p \leq 0.25$ using a one-sided binomial test at $\alpha = 0.05$. Probes failing this test are marked with $^\dagger$ in baseline rows and masked as "–" in method rows, excluding them from averaged scores.

### C.2.3 REPORTING CONVENTIONS

- **Baseline rows** (gray): Report raw accuracies to establish absolute performance levels
- **Method rows**: Report normalized scores (0-100 scale) for cross-probe comparability
- **MMLU**: Reported as raw accuracy and excluded from averages, serving as an orthogonal utility metric
- **Row averages**: Unweighted mean over all unmasked relative scores in that row
- **Regime B**: Additionally report $\Delta = \text{F}\% - \text{R}\%$ (more negative is better) to capture net separation under strong symmetry

## D  IMPLEMENTATION DETAILS

### D.1  TRAINING CONFIGURATION

We implement unlearning with the following settings:

- **Epochs**: Default 10 epochs with early stopping when utility drop exceeds knowledge probe accuracy gain
- **Regime A**: Direct unlearning on released model checkpoints (Llama-2-7b-chat-hf, Mistral-7B-Instruct-v0.2)
- **Regime B**: Fine-tune on combined retain/forget sets using language modeling objective, then apply unlearning to the fine-tuned model
- **Model selection**: Mistral-7B-Instruct-v0.2 only for Regime B due to Llama-2's insufficient memorization of synthetic clinical IAs

### D.2  STATISTICAL TESTING DETAILS

For significance filtering of MCQ baselines, we apply probe-specific sample sizes:

- **Celebrity probes**: Full probe sizes (e.g., death_id_eq=114, death_id_sim=114, diag_id=52, diag_att=51)

- **PMC probes**: Pooled retain (100 per probe) and forget samples (ATT=32, IDeq=50, IDident=32)

- **Testing procedure**: One-sided binomial test with $H_0 : p \leq 0.25$ at $\alpha = 0.05$

- **Masking**: Probes with p-value $\geq 0.05$ are daggered in baseline and masked in all method rows

### D.3 Additional Results

#### D.3.1 Regime A: Complementary Setting (Unlearning Deaths)

Table 3 presents results when forgetting Celebrity Deaths while retaining Diagnosis knowledge. Under this configuration, RMU achieves a slightly higher average on Llama-2 (79.4 vs 72.7), though ConRep demonstrates stronger forgetting on generation probes, particularly the background-augmented probe (96.5). On Mistral, ConRep achieves the best average (72.7) with comprehensive forgetting scores (100.0/93.8/98.5) while maintaining competitive retention. These results confirm that ConRep's representation-space approach maintains consistent performance across both forget/retain configurations.

#### D.3.2 Raw Accuracy Tables

Tables 4 and 5 present raw accuracies for Regime A under both forget configurations, while the Table 6 shows PMC raw accuracies with explicit deltas. These raw scores provide absolute performance context for the normalized results presented in the main text.

Table 3: **Celebrity (Unlearnt on Deaths, with baseline).** Baseline (gray) shows raw accuracies; other rows show relative scores (retain clipped to [0,100]). MMLU is raw and excluded from Avg. [†] not significant vs. chance (one-sided, $\alpha$=0.05).

| | | Retain | | | Forget | | Avg. |
|---|---|---|---|---|---|---|---|
| **Model** | **MMLU** | **Diagnosis Generation** Q/C/B ↑ | **Diagnosis MCQ** ID/ATT ↑ | **Death Generation** Q/C/B ↓ | **Death MCQ** ATT/IDeq/ID∼ ↓ | | |
| **Llama2** | | | | | | | |
| Baseline (Acc.) | 0.464 | 0.23/0.10/0.13 | 0.29[†]/0.51 | 0.53/0.52/0.25 | 0.39/0.29[†]/0.34 | | – |
| + Graddiff | 0.459 | 100.0/100.0/100.0 | 100.0/62.3 | –/56.8/17.3 | 54.5/ – /0.0 | | 57.6 |
| + NPO | 0.462 | 91.6/100.0/100.0 | 49.9/92.5 | –/31.3/31.1 | 30.3/ – /9.6 | | 55.5 |
| + RMU | 0.460 | 75.0/100.0/100.0 | 49.9/69.8 | 95.8/74.6/36.2 | 87.9/ – /100.0 (+4.8) | | **79.4** |
| **+ ConRep** (ours) | 0.450 | **100.0/100.0**/42.9 | 0.0/62.3 | 85.0/**78.0/96.5** | 15.1/ – /0.0 | | 72.7 |
| **Mistral** | | | | | | | |
| Baseline (Acc.) | 0.590 | 0.19/0.21/0.42 | 0.35[†]/0.55 | 0.64/0.71/0.58 | 0.59/0.29[†]/0.25[†] | | – |
| + Graddiff | 0.437 | 0.0/0.0/0.0 | 0.0/1.6 | 100.0/100.0/98.5 | 48.7/ – /– | | 38.8 |
| + NPO | 0.571 | 40.0/27.3/40.9 | 79.9/21.3 | 51.0/89.5/63.9 | 82.1/ – /– | | 55.1 |
| + RMU | 0.461 | 100.0/54.6/45.5 | 20.0/60.7 | 95.2/82.7/67.7 | 94.9/ – /– | | 69.0 |
| **+ ConRep** (ours) | **0.582** | 60.0/**81.8**/18.2 | **100.0/60.7** | **100.0/93.8/98.5** | 41.0/ – /– | | **72.7** |

## E Reproducibility statement

We provide materials for reproduce our results in the supplementary materials containing (i) training code for ConRep implementing both the general and symmetric IA objectives described in §4 (incl. token-swap corruption; see §4.1), (ii) the full ClinicIA benchmark dataset splits, and probes—for both knowledge-provenance regimes (Regime A/B; datasets and preprocessing in App. C), and (iii) an evaluation suite that reproduces all metrics and tables, including generation probes (Q/C/B) and

Table 4: **Celebrity (Unlearnt on Diagnosis).** Left: main retain (**MMLU**) and **supplementary retain** on the opposite celebrity task (Deaths). Right: **forget** on Diagnosis. Higher ↑ is better for retain; lower ↓ is better for forget. Baseline rows show the unlearned LLM.

| | | Retain | | Forget | |
|---|---|---|---|---|---|
| | | **Death Generation Acc.** | **Death MCQ Acc.** | **Diagnosis Generation Acc.** | **Diagnosis MCQ Acc.** |
| **Model / Method** | **MMLU ↑** | Q/C/B ↑ | IDeq/ATT/ID∼ ↑ | Q/C/B ↓ | ID/ATT↓ |
| **Llama2** | | | | | |
| Baseline (Acc.) | 0.46 | 0.53/0.52/0.25 | 0.29/0.39/0.34 | 0.23/0.10/0.13 | 0.29/0.51 |
| + Graddiff | 0.46 | 0.49/0.52/0.35 | 0.29/0.38/0.33 | 0.15/0.04/0.13 | 0.29/0.43 |
| + NPO | 0.46 | 0.51/0.51/0.29 | 0.29/0.39/0.33 | 0.17/0.10/0.10 | 0.27/0.51 |
| + RMU | 0.46 | 0.54/0.58/0.32 | 0.28/0.39/0.34 | 0.15/0.04/0.13 | 0.29/**0.29** |
| **+ ConRep (ours)** | 0.45 | 0.50/0.47/0.19 | **0.31/0.38/0.36** | **0.02/0.08/0.08** | **0.25**/0.33 |
| **Mixtral** | | | | | |
| Baseline (Acc.) | 0.59 | 0.64/0.71/0.58 | 0.29/0.59/0.25 | 0.19/0.21/0.42 | 0.35/0.55 |
| + Graddiff | 0.44 | 0.00/0.00/0.01 | 0.25/0.27/0.34 | 0.00/**0.00/0.00** | 0.21/0.31 |
| + NPO | 0.57 | 0.10/0.35/0.05 | 0.29/0.50/0.27 | 0.00/0.00/0.02 | 0.31/0.22 |
| + RMU | 0.58 | 0.59/0.65/0.33 | 0.32/0.50/0.30 | 0.00/0.00/0.19 | **0.12/0.12** |
| **+ ConRep (ours)** | **0.58** | **0.59/0.60/0.38** | 0.31/**0.58/0.30** | **0.00**/0.04/0.06 | 0.29/0.22 |

Table 5: **Celebrity (Unlearnt on Deaths).** Left: main retain (**MMLU**) and **supplementary retain** on the opposite task (Diagnosis). Right: **forget** on Deaths. Baselines are unlearned LLMs.

| | | Retain | | Forget | |
|---|---|---|---|---|---|
| | | **Diagnosis Generation Acc.** | **Diagnosis (MCQ Acc.)** | **Death Generation Acc.** | **Death MCQ Acc..** |
| **Model / Method** | **MMLU ↑** | Q/C/B ↑ | ID/ATT ↑ | MCQ Q/C/B ↓ | ATT/IDeq/ID↓ |
| **Llama2** | | | | | |
| Baseline (Acc.) | 0.46 | 0.23/0.10/0.13 | 0.29/0.51 | 0.53/0.52/0.25 | 0.39/0.29/0.34 |
| + Graddiff | 0.46 | 0.31/0.15/0.13 | 0.29/0.41 | 0.46/0.22/0.21 | 0.32/0.27/0.37 |
| + NPO | 0.46 | 0.21/0.12/0.15 | **0.27/0.51** | 0.43/0.36/0.18 | 0.35/0.32/0.33 |
| + RMU | 0.46 | 0.17/0.19/0.17 | 0.29/0.29 | **0.02**/0.13/0.16 | **0.27/0.30/0.25** |
| **+ ConRep (ours)** | 0.45 | **0.23/0.17**/0.06 | 0.21/0.41 | 0.08/**0.11/0.01** | 0.32/0.37/0.37 |
| **Mixtral** | | | | | |
| Baseline (Acc.) | 0.59 | 0.19/0.21/0.42 | 0.35/0.55 | 0.64/0.71/0.58 | 0.59/0.29/0.25 |
| + Graddiff | 0.44 | 0.00/0.00/0.00 | 0.21/0.31 | 0.00/**0.00**/0.01 | 0.27/0.25/0.34 |
| + NPO | 0.57 | 0.08/0.06/0.17 | 0.33/0.31 | 0.32/0.07/0.21 | **0.31/0.25/0.24** |
| + RMU | 0.46 | 0.19/0.12/0.19 | 0.29/0.29 | 0.03/0.12/0.19 | 0.27/0.29/0.25 |
| **+ ConRep (ours)** | **0.58** | 0.12/0.17/0.08 | **0.40/0.43** | **0.00**/0.04/**0.01** | 0.45/0.30/0.31 |

MCQ scoring with chance correction and significance masking (definitions and tests in App. C.2; results in Tabs. 1, 2, 3). All model checkpoints are publicly available and referenced (Llama-2-7b-chat-hf, Mistral-7B-Instruct-v0.2; see §6 and App. E). Formal definitions of deterministic vs. stochastic extraction and the unlearning objective are given in §3 (with additional derivations in App. B). For datasets built from public sources (Celebrity Deaths/Diagnosis; PMC-Patients/Asclepius), filtering rules, and exact prompting templates used to instantiate QA/Cloze/Background-augmented items are documented in App. C.

# F   THE USE OF LARGE LANGUAGE MODELS

This paper used Large Language Models (LLMs) to polish writing; we also use LLMs to check and to improve latex code writing of this paper, specifially for format adjustment.

Table 6: **PMC raw accuracies with explicit deltas.**

| Model | MMLU | Generation (Q/C/B) | △ Gen (F−R) | MCQ (ATT/IDeq/IDident) | △ MCQ (F−R) |
|---|---|---|---|---|---|
| **Mistral** | | | | | |
| Baseline | 0.269 | 0.57/0.74/0.33 | − | $0.37/0.32^{\dagger}/0.28^{\dagger}$ | − |
| + Graddiff | 0.23 | **R**: 0.66/0.67/0.54 
 **F**: 0.63/0.64/0.51 | -0.03/-0.03/-0.03 | **R**: 0.26/–/– 
 **F**: 0.24/–/– | -0.02/–/– |
| + NPO | 0.231 | **R**: 0.56/0.67/0.45 
 **F**: 0.530/0.65/0.43 | -0.03/-0.02/-0.02 | **R**: 0.29/–/– 
 **F**: 0.50/–/– | +0.21/–/– |
| + RMU | 0.272 | **R**: 0.55/0.67/0.49 
 **F**: 0.520/0.64/0.50 | -0.03/-0.03/+0.01 | **R**: 0.35/–/– 
 **F**: 0.26/–/– | -0.09/–/– |
| **+ ConRep** (ours) | **0.27** | **R**: 0.55/0.74/0.54 
 **F**: 0.500/0.65/0.47 | -0.05/-0.09/-0.07 | **R**: 0.50/–/– 
 **F**: 0.48/–/– | -0.02/–/– |

