# OpenReview forum: "Towards Unlearning Beyond Textual Expressions for LLMs"
_ICLR.cc/2026/Conference — ICLR 2026 Conference Withdrawn Submission_

### Official Review · Reviewer_K93C · 2025-10-28

**Soundness:** 2
**Presentation:** 3
**Contribution:** 2
**Rating:** 4
**Confidence:** 5

**Summary:**

The paper argues that existing unlearning methods optimize behavior on fixed textual probes and thus "forget" only surface forms. It formalizes knowledge extraction as a stochastic event marginalized over a latent distribution of admissible prompts and reframes unlearning as minimizing this distributional extraction risk. To make this tractable, it proposes ConRep, a representation-space contrastive objective. The authors also introduce ClinicIA, a benchmark for identifier–attribute (IA) unlearning with generation and MCQ probes.

**Strengths:**

The method grounded in representation geometry is well motivated. The authors also propose ClinicIA, which comprises multiple potential vulnerabilities of "surface" unlearning. Empirical results show improved unlearning effectiveness while largely preserving MMLU performance.

**Weaknesses:**

1. While stochastic extraction is an interesting idea, the paper lacks formal analysis or guarantees that the proposed representation-space surrogate (contrastive losses + token-swap) truly minimizes marginalized extraction risk beyond intuition. A tighter theoretical connection or risk bound would strengthen the claim.

2. The authors distinguish ConRef from coarse-grained unlearning methods like RMU in "Related Studies" section, but a comparison with fine-grained unlearning works (e.g., [1]) is missing. Any discussions and benchmarking would better contextualize ConRef’s advantage both as a method and its performance.

3. ClinicIA is a useful diagnostic benchmark but limited in topical and structural diversity. To validate ConRef’s generality, experiments on standard datasets and metrics (ROUGE-1 on datasets such as TOFU and PISTOL) are needed.

4. It would be valuable to test whether unlearned information can still be probed from earlier layers (i.e., bypass the representation-engineering done with the final layer). Also given the current implementation, will quantization-based attack [2] recover information? Such tests would clarify the robustness of the proposed parameter update mechanism.

[1] LLM Unlearning via Neural Activation Redirection
[2] Catastrophic Failure of LLM Unlearning via Quantization

**Questions:**

See above

---

### Official Review · Reviewer_e5Ch · 2025-10-29

**Soundness:** 2
**Presentation:** 2
**Contribution:** 2
**Rating:** 2
**Confidence:** 5

**Summary:**

The paper introduces ConRep, a contrastive representation-based unlearning method that operates in the latent space to achieve deeper forgetting. It also proposes ClinicIA, a new benchmark designed to evaluate the robustness of unlearning methods against diverse probing variants. The authors claim that ConRep achieves better performance compared to GradDiff, NPO, and RMU methods.

**Strengths:**

The paper aims to tackle the important and timely challenge of achieving deeper unlearning. In addition, the authors introduce the ClinicIA dataset, which provides a valuable new evaluation dimension by incorporating multiple probing variants to assess unlearning robustness.

**Weaknesses:**

The core idea of ConRep is relatively straightforward, and the evaluation setup is limited in scope, relying on outdated models (e,g., need to evaluate on models such as LLaMA3, Qwen) and baseline methods (e.g., DPO, and stronger and more recent baselines need to be benchmarked) to better demonstrate generality and performance gain. Additionally, normalization with clipping can obscure absolute differences and exaggerate small deltas when baselines are weak. Also from the few raw tables in the appendix, observed gains appear modest and inconsistent.

Additionally, the loss jointly optimizes representation and token spaces, which complicates the claim of being “representation-grounded.” It also results in large number of hyperparameters (ρ, τ_f/τ_r, w, γ, λ, etc.) raises concerns about its potential for real deployment and stability. There's a lack of ablation study on how sensitive are unlearning results to the choice of these hyperparameters?

While the proposed ClinicIA benchmark is valuable, the method’s applicability to established datasets such as TOFU should also be shown. Raw scores should be shown more clearly to interpret the significance of performance improvement.

**Questions:**

N/A

---

### Official Review · Reviewer_c58F · 2025-11-01

**Soundness:** 2
**Presentation:** 2
**Contribution:** 3
**Rating:** 2
**Confidence:** 3

**Summary:**

The paper studies the problem of knowledge unlearning, where the goal is to make a specific piece of knowledge unextractable from an LLM through any textual query after the unlearning process. Prior approaches typically convert the target knowledge into a single textual description and perform unlearning based on that description. However, this does not guarantee that the knowledge cannot be recovered via paraphrased or semantically equivalent queries. To address this limitation, the paper proposes a new unlearning method that introduces a contrastive loss defined in the embedding space of the LLM’s pooled last-layer representations. Empirical evaluations on a constructed benchmark show that the proposed method consistently outperforms existing approaches.

**Strengths:**

1. The challenge studied by the paper is important in the studying of unlearning. If the goal is to make the target knowledge piece unextractable, it is challenging to make the unlearning effectively against all forms of extractions.
2. The proposed method is novel to my best knowledge. The proposed loss is built on the pooled last-layer embeddings. The loss encourage the embedding internally collapes among the knowledge inside the retain set and the forget set respectively and push away the embeddings between the retain and forget sets.

**Weaknesses:**

1. An important related study, [1], is not discussed, which makes it difficult to fully assess the novelty of the problem formulation and the significance of the results. Specifically, [1] investigates knowledge extraction through paraphrased texts within the forget set, which is similar to the problem set-up of this paper. Further, [1] evaluates several unlearning methods for this setting. Without a direct comparison or discussion of how the proposed method differs from or improves upon those in [1], it remains unclear what conceptual or empirical advancements this paper contributes beyond prior work.
2. If my understanding of the setup is correct, the three baseline methods are evaluated without incorporating augmented texts. Since it would be straightforward to extend these baselines to use the same augmented data, such a comparison would more clearly check the benefits of the proposed loss function and demonstrate its effectiveness beyond data-level augmentation.
3. The formulation of the loss function requires further clarification. For example, why do the positive pairs for a retained knowledge piece include other distinct knowledge pieces within the retain set? A similar question arises for the construction of the forget loss.
4. Several terms and phrases are vague or insufficiently explained, making some parts of the paper difficult to interpret. Here are some examples:
    - The explanation of “symmetry” (line 348) is unclear.
    - The term “injection” (lines 355–356) is not defined.
    - The phrase “complementary knowledge-provenance regimes” (lines 317–318) is difficult to understand and should be clarified.
    - In lines 349–350, the phrase “deeply ingrained knowledge” is ambiguous — what is meant by “deeply,” and how should “ingrained knowledge” be interpreted in this context?
    - The statement “Deep embeddings expose surface-level brittleness” (lines 434–435) also needs explanation — what do “deep” and “surface-level” specifically refer to here?

[1] Patil, Vaidehi, Peter Hase, and Mohit Bansal. "Can Sensitive Information Be Deleted From LLMs? Objectives for Defending Against Extraction Attacks." The Twelfth International Conference on Learning Representations.

**Questions:**

Please check the Weaknesses section.

---

### Note · Authors · 2025-11-28

I have read and agree with the venue's withdrawal policy on behalf of myself and my co-authors.